# New geographic model of care to manage the post-COVID-19 elective surgery aftershock in England: a retrospective observational study

Jonathan Clarke ,[1] Alice Murray,[2] Sheraz Rehan Markar ,[2] Mauricio Barahona ,[1] James Kinross,[2] on behalf of The PanSurg Collaborative

► Prepublication history and additional materials for this paper is available online. To view these files, please visit the journal online (http://dx.doi.org/10.1136/bmjopen-2020-042392).

JC and AM contributed equally.

¹Department of Mathematics, Imperial College of Science, Technology and Medicine, London, UK
²Department of Surgery and Cancer, Imperial College of Science, Technology and Medicine, London, UK

**Correspondence to**
Dr Jonathan Clarke;
j.clarke@imperial.ac.uk

## ABSTRACT

**Objectives** The suspension of elective surgery during the COVID-19 pandemic is unprecedented and has resulted in record volumes of patients waiting for operations. Novel approaches that maximise capacity and efficiency of surgical care are urgently required. This study applies Markov multiscale community detection (MMCD), an unsupervised graph-based clustering framework, to identify new surgical care models based on pooled waiting-lists delivered across an expanded network of surgical providers.

**Design** Retrospective observational study using Hospital Episode Statistics.

**Setting** Public and private hospitals providing surgical care to National Health Service (NHS) patients in England.

**Participants** All adult patients resident in England undergoing NHS-funded planned surgical procedures between 1 April 2017 and 31 March 2018.

**Main outcome measures** The identification of the most common planned surgical procedures in England (high-volume procedures (HVP)) and proportion of low, medium and high-risk patients undergoing each HVP. The mapping of hospitals providing surgical care onto optimised groupings based on patient usage data.

**Results** A total of 7 811 891 planned operations were identified in 4 284 925 adults during the 1-year period of our study. The 28 most common surgical procedures accounted for a combined 3 907 474 operations (50.0% of the total). 2 412 613 (61.7%) of these most common procedures involved 'low risk' patients. Patients travelled an average of 11.3 km for these procedures. Based on the data, MMCD partitioned England into 45, 16 and 7 mutually exclusive and collectively exhaustive natural surgical communities of increasing coarseness. The coarser partitions into 16 and seven surgical communities were shown to be associated with balanced supply and demand for surgical care within communities.

**Conclusions** Pooled waiting-lists for low-risk elective procedures and patients across integrated, expanded natural surgical community networks have the potential to increase efficiency by innovatively flexing existing supply to better match demand.

## Strengths and limitations of this study

► The COVID-19 pandemic has significantly disrupted the provision of planned surgical care in hospitals across the world. Addressing the accumulated backlog of cases requires a new model of care whereby procedures are carried out at pace, while also responding to the dynamic risk of further COVID-19 outbreaks.

► This study uses national, retrospective hospital administrative data relating to 7.8 million interventional procedures in 4.2 million adults.

► Markov multiscale community detection, an unsupervised network clustering technique, is applied to understand how providers of surgical care may collaborate with one another based on prior patterns of surgical care delivery.

► The relative imbalances in supply and demand for surgical care within the identified surgical communities are quantified in order to determine the potential applicability of different scales of collaboration between care providers.

► While this study advances the potential role of collaboration between surgical centres to address the surgical backlog resulting from COVID-19, it does not address issues relating to local financial or logistical barriers to implementation of such a strategy.

## INTRODUCTION

The COVID-19 pandemic put a global halt to the majority of elective surgery in order to manage the surge in patients requiring acute hospital services and intensive treatment unit care.[1–4] It has been estimated that 28 million elective operations worldwide have been cancelled or postponed due to the pandemic.[5] Although the focus of public health organisations globally was rightly mounting an effective emergency response to the COVID-19 pandemic, the surgical 'aftershock' will therefore be unprecedented and yet to be fully appreciated. Millions of patients in the UK are already waiting for treatment

and numbers increase daily as the diversion of resources continues.[6] Elective surgical services are gradually being reintroduced, aiming to treat waiting patients without risking the spread of COVID-19. Management strategies in the UK are currently focused on undertaking life-saving cancer operations in 'clean' COVID-free hospitals or in hospital sites away from the acute care sites where COVID-19 is more prevalent.[7 8] An immediate response to 'catch up' and clear case load will need to be undertaken, as well as adjusting to a 'new normal'.

Waiting-list numbers vary widely across the country and waiting times have increased in recent years.[9] To add complexity, there is also regional variation in the number of COVID-19 infections and burden of COVID-related workload.[10 11] Therefore, in order to respond to the needs of a particular population, dynamic, flexible and regional solutions will be required to balance the reintroduction of services with careful COVID-19 management.

Flexibility in the location where care is provided, according to patients' clinical needs, has the potential to better match supply of services where there is appropriate demand. Patients can be treated more promptly if surgeons, hospitals and hospital delivery systems work together across provider networks, managing a centrally pooled workload. While some patients will need to be treated at specific locations (particularly high-risk patients or those requiring complex cancer care), there are other less complex procedures that could feasibly be performed by a range of qualified providers for patients who are able to travel.[12]

As the National Health Service (NHS) in England moves towards greater integration, there is an opportunity to break down arbitrary geographic boundaries and funding barriers, and bring together multiple providers of surgical care into 'surgical communities'. In such configurations, hospitals share a centrally managed waiting-list for routine surgical procedures, and patients may receive surgery at any centre within the community of providers with the capacity to do so. There is a precedent for this approach, as a similar scheme was successfully piloted on a small scale in London.[13] Pooling available capacity between communities of surgical care providers may enable the efficient use of their collective available resources.

In this study we explore the potential of using flexible locations of care as a strategy to manage waiting-lists. First, we categorise the types of elective procedures and eligible patients into groups that would be amenable to undergoing surgery in any suitable location. Second, we identify from patient data existing community networks of surgical providers ('surgical communities') that collectively provide planned surgical care to similar geographic patient populations. Third, we map these surgical communities against existing organisational configurations and model the effect on supply and demand when patients travel further for care.

## METHODS

All planned inpatient admissions to hospitals in England involving a surgical procedure were identified for adults resident in England from Hospital Episode Statistics from 1 April 2017 to 31 March 2018. NHS-funded procedures conducted in non-NHS hospitals were included. For each admission, the first operative day was defined as the first day within an admission in which a surgical procedure was recorded. Procedures performed after the first operative day were excluded from the analysis. Where multiple procedures were performed on the first operative day, all of those procedures were counted to capture the fullest reliable representation of planned surgical activity. Inclusion of procedures after the first operative day is likely to include unplanned operations arising from surgical complications which are not identifiable as unplanned procedures in the data available.

All procedure codes describing diagnostic imaging, testing or rehabilitation (OPCS Classification of Intervention and Procedures version-4 codes beginning with U), the method of a procedure (Y) and site of a procedure (Z) were removed in addition to miscellaneous operations (X).[14] Procedures involving the concurrent extraction of a lens (C71) and insertion of a prosthetic lens (C75) were treated as a single procedure. Lower gastrointestinal (GI) diagnostic and therapeutic endoscopies frequently occurred concurrently or under codes with similar descriptions and were therefore grouped together. Conversely, diagnostic upper GI endoscopy (G45) was far more common than therapeutic endoscopies and was therefore treated separately.

### Classification of operative risk

For each procedure, the age of the patient at the time of surgery was extracted. The modified Charlson comorbidity score of each patient was determined based on the presence of International Classification of Diseases 10th Revision diagnosis codes extracted from their operative admission and all other recorded admissions to hospital for each patient in the 6 months prior to surgery.[15] Patients were then classified according to low, medium or high risk (for potential morbidity and mortality) by virtue of their age and Charlson score (table 1).

### Identification of high-volume procedures

The total number of procedures performed for each three-digit OPCS-4 code was calculated and sorted in

**Table 1** Classification of low, medium and high-risk patients based on age and Charlson score

| | | Charlson score | | |
|---|---|---|---|---|
| | | **0** | **1–2** | **3+** |
| Age | <60 | Low | Low | Medium |
| | 60–74 | Low | Medium | High |
| | 75+ | Medium | High | High |

descending order by volume. Those top procedures collectively accounting for more than 50% of the overall number of procedures were selected, and hereafter referred to as 'High Volume Procedures' (HVP).

## Identification of hospital sites

The site in which a procedure was performed was identified from the SITETRET code of its associated admission. The postcodes of all sites in which procedures were performed were extracted from the site-level Estates Returns Information Collection.[16] Postcodes were converted to latitude and longitude coordinates. For all sites, the straight-line distance between all sites was calculated using the haversine formula.[17] Where sites were within 1 km of one another, they were treated as a single merged site under the code and coordinates of the highest volume provider.

## Calculation of distance travelled for surgery

For each patient, the approximate location of their home was determined using the coordinates of the population-weighted centroid of their lower layer super output area (LSOA) of residence.[18] LSOAs are mutually exclusive, collectively exhaustive geographic census divisions defined by the UK Office for National Statistics, of which there are 32 844 in England, with a mean population of 1704 people, and is therefore similar in scale to census block groups in the USA. The straight-line distance between the population-weighted centroid of the LSOA of residence of the patient and the site in which the procedure was performed was calculated according to the haversine formula.

For each HVP, the total number of procedures performed was calculated. The number of patients classified as low, medium and high risk was calculated, along with the total number of sites undertaking the procedure and the average distance travelled for surgery. For each HVP, the total number of procedures performed by each site was calculated. To exclude providers who rarely perform a procedure, the highest volume providers who collectively accounted for 99% of procedures were identified and classified as providers of the HVP.

## Identification of surgical communities

The proportion of patients presenting from each LSOA in England to each regular provider site for an HVP was calculated and a normalised cosine similarity matrix of LSOAs was computed (Equation 1).

$$\text{similarity}_{AB} = \frac{\sum_{i=1}^{n} A_i B_i}{\sqrt{\sum_{i=1}^{n} A_i^2} \sqrt{\sum_{i=1}^{n} B_i^2}} \qquad (1)$$

## Equation 1

Calculation of cosine similarity between LSOAs. $A_i$ is the proportion of patients presenting to hospital site $i$ resident in LSOA A; $B_i$ is the proportion of patients presenting to hospital site $i$ resident in LSOA B; and $n$ is the total number of hospital sites in the data set.

This matrix quantifies the similarity of patterns of presentation for HVPs between all LSOAs in England. It can be understood as the adjacency matrix of a dense, weighted network connecting LSOAs to one another according to the similarity in their patterns of presentation to hospital for HVPs.[19] This network was sparsened using the relaxed minimum spanning tree technique, a method used elsewhere in applied network science to sparsen a dense, inhomogeneous network to preserve both local and global connectivity within a network.[20 21] This sparsened network was subsequently partitioned using Markov multiscale community detection (MMCD) to produce partitions of the LSOAs according to shared patterns of presentation to hospital sites for HVPs.[22 23]

## Description of surgical communities

The total number of procedures performed in each surgical community and the total number of hospital sites were calculated. For each sustainability and transformation partnership (STP-NHS organisational divisions of England into 44 regions responsible for developing local integration between primary and secondary care providers), the effective number of surgical communities active within its boundary was calculated using the equivalent market size (the reciprocal of the Herfindahl-Hirschman Index of market concentration) (Equation 2).[24]

$$\text{EMS}_i = 1 / \sum_{j=1}^{N} s_{ij}^2 \qquad (2)$$

## Equation 2

The equivalent market size of $STP_i$. Here, $s_{ij}$ is the proportion of LSOAs in $STP_i$ contained within surgical community $j$, and $N$ is the number of surgical communities in the partition.

## Calculation of the balance between supply and demand within surgical communities

Surgical communities were modelled as self-contained subdivisions of England containing LSOAs contributing cases requiring surgery (demand) and hospitals providing finite surgical capacity for those services (supply).[25] In this configuration, surgical procedures for patients resident within a surgical community would be performed at a hospital site spatially located within the same surgical community. Within each surgical community, surgical demand was calculated as the total number of HVP cases performed for patients resident in LSOAs within the surgical community. Supply was calculated as the total number of HVP cases performed by sites located within the geographic boundary of the surgical community. The supply-demand mismatch was calculated as the percentage difference between supply and demand for each community. The median of the absolute value of the supply-demand mismatch was determined.

## Patient and public involvement

We did not directly include patient and public involvement in this study, but the database used in the study was

**Table 2** The 28 procedures accounting for more than half of all elective surgical activities in England

| Procedure | Total number of cases | Patient risk | | | Mean distance travelled (km) |
|---|---|---|---|---|---|
| | | Low risk (%) | Medium risk (%) | High risk (%) | |
| Lower GI endoscopy | 937 616 | 74.8 | 17.9 | 7.3 | 9.8 |
| Upper GI endoscopy | 650 133 | 66.9 | 22.1 | 10.9 | 9.4 |
| Lens extraction+ replacement | 395 445 | 33.5 | 46.5 | 20.0 | 10.9 |
| Excision of skin lesion | 215 608 | 55.0 | 29.5 | 15.5 | 12.7 |
| Injection/aspiration joint | 142 562 | 71.6 | 20.9 | 7.5 | 12.6 |
| Vitrectomy | 132 938 | 39.9 | 44.1 | 16.1 | 13.2 |
| Cystoscopy | 130 114 | 56.4 | 26.2 | 17.4 | 11.8 |
| Insertion of central venous catheter | 109 864 | 24.3 | 38.3 | 37.4 | 14.0 |
| Coronary angiography | 105 620 | 56.2 | 30.0 | 13.8 | 13.9 |
| Dental extraction | 101 435 | 91.6 | 5.8 | 2.5 | 11.5 |
| Knee replacement | 78 773 | 53.3 | 34.4 | 12.3 | 13.4 |
| Bladder catheterisation or irrigation | 71 552 | 42.5 | 32.7 | 24.8 | 12.8 |
| Injection to bladder | 67 167 | 34.3 | 29.8 | 35.9 | 11.5 |
| Spinal facet joint injection | 64 154 | 70.4 | 21.9 | 7.7 | 14.0 |
| Cholecystectomy | 61 790 | 80.5 | 13.8 | 5.7 | 11.8 |
| Lymph node biopsy | 60 674 | 34.8 | 34.4 | 30.8 | 14.9 |
| Epidural or spinal injection | 60 656 | 69.2 | 22.6 | 8.1 | 12.9 |
| Inguinal hernia repair | 58 943 | 72.6 | 19.9 | 7.5 | 10.9 |
| Spinal nerve root injection | 58 212 | 77.0 | 17.5 | 5.5 | 16.2 |
| Knee meniscectomy/ meniscal repair | 57 871 | 93.2 | 5.9 | 0.8 | 12.4 |
| Hysteroscopy | 52 360 | 90.9 | 6.4 | 2.7 | 9.9 |
| Carpal tunnel release | 48 245 | 70.7 | 22.2 | 7.1 | 11.0 |
| Application/ removal of internal fixation of bone | 46 771 | 84.6 | 11.9 | 3.5 | 15.7 |
| Dental clearance | 43 463 | 82.3 | 11.7 | 5.9 | 11.1 |
| Partial breast excision | 41 827 | 50.1 | 31.4 | 18.5 | 11.5 |
| Bone marrow biopsy | 38 369 | 39.8 | 34.8 | 25.5 | 15.6 |
| Primary joint resurfacing | 37 854 | 59.3 | 30.1 | 10.5 | 14.0 |
| Cystoscopy+ resection of bladder lesion | 37 458 | 17.7 | 29.8 | 52.5 | 11.2 |

The proportion of patients classified as low, medium and high risk according to table 1 is shown, along with the mean distance travelled from a patient's LSOA of residence to the hospital site in which the procedure is performed.
GI, gastrointestinal; LSOA, lower layer super output area.

released following review by a panel including patient representatives.

## RESULTS

A total of 7 811 891 planned interventional procedures corresponding to 5 718 031 admissions involving 4 284 925 adult patients resident in England from 1 April 2017 to 31 March 2018 were identified. These procedures were performed at 530 NHS hospital sites and 162 different private provider sites. A total of 1210 different three-digit OPCS codes were used.

Twenty-eight types of procedure in table 2 accounted for 3 907 474 operations, over half of all planned surgical procedures during the study period. These are denoted as HVPs. Of these HVPs, 3 553 649 (90.9%) were performed in an NHS site, while 353 825 (9.9%) were performed in a non-NHS site. Collectively, diagnostic or therapeutic upper and lower GI endoscopy accounted for 1.6 million procedures (20.3%). On average, procedures were performed on patients aged 61.4 years (SD=16.7 years). A total of 2 636 559 procedures were performed on patients with a Charlson comorbidity score of 0 (67.5%), while 997 765 procedures were performed on patients with a Charlson score of 1 or 2 (25.5%) and 273 150 procedures were performed on patients with a Charlson score of 3 or more (7.0%).

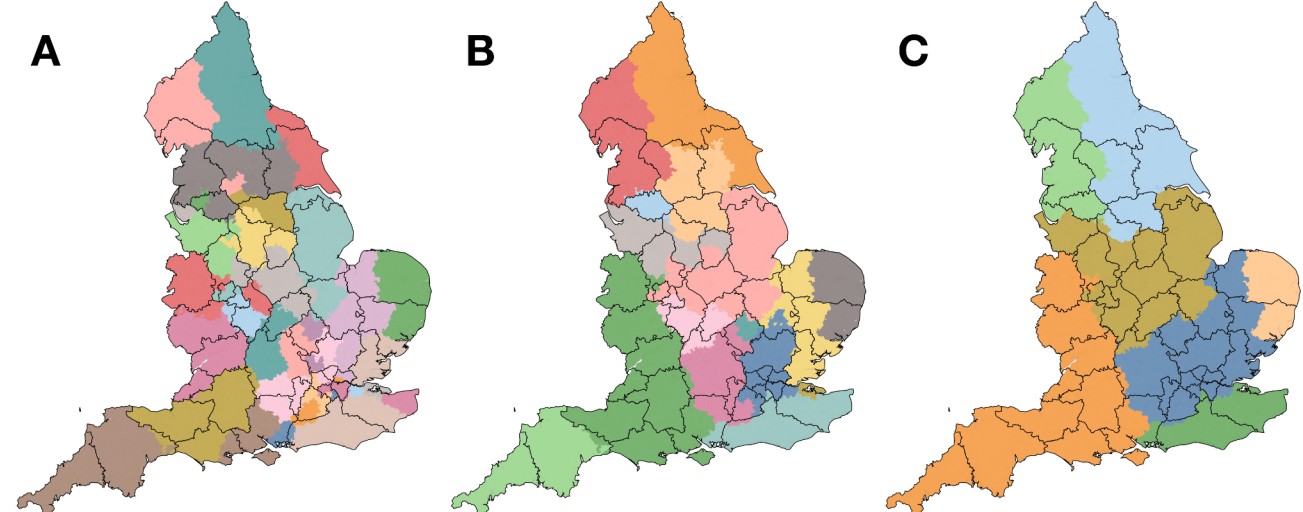

**Figure 1** Division of England into 45 (panel A), 16 (panel B) and 7 (panel C) surgical communities (in colour), according to Markov stability. Sustainability and transformation partnership (STP) boundaries are overlaid (black lines).

The mean distance travelled from a patient's residence to hospital for surgery was on average 11.3 km. Mean distances for the 28 HVPs ranged from 9.4 km for upper GI endoscopy to 16.2 km for spinal nerve root injection. A total of 2 412 613 (61.7%) HVPs were performed in 'low risk' patients, 988 067 (25.3%) in 'medium risk' patients and 506 794 (13.0%) in 'high risk' patients. The proportion of procedures being performed on 'high risk' patients ranged from 1% for meniscal procedures to 52% for cystoscopy and resection of bladder lesions. In 22 out of 28 HVPs, more than 80% of patients were classified as 'low' or 'medium' risk.

MMCD identified (see online supplemental figure 1) three robust community conformations of LSOAs consisting of 45 (partition A), 16 (partition B) and 7 (partition C) surgical communities (figure 1). Stable spatial motifs are observed across the three partitions.

Overlaid STP boundaries show variable agreement with surgical communities (figure 1). Lower agreement is observed, for example, in East Anglia, where surgical communities consistently partition in 'north-south' direction, while the STP boundary runs 'east to west'. Close agreement can be seen in Cornwall, where STPs are adjoining, based around surgical communities. The Hampshire and Isle of Wight STP, in the south of England, remains divided between more than three surgical communities in partition C.

The median number of HVP cases performed in each community ranges from 78 998 in the finest partition (A) to 574 403 in the coarsest partition (C) (table 3). In partition A, the median number of surgical sites per community is 9, with an IQR from 9 to 17. In partition B, the median number of surgical sites per community is 25, with an IQR of 19–44, while in partition C, a median of 84 surgical sites are present per community, with an IQR of 56–98. In partition A, STPs involved a median of 1.7 surgical communities, compared with 1.1 for partition B and 1.0 for partition C. Only the Hampshire and the Isle of Wight STP remains divided between more than three surgical communities in partition C (figure 2).

### Supply and demand relationships within surgical communities

In partition A, median absolute percentage difference between supply and demand for HVPs within surgical communities is 5.1%. Twelve communities (27%) had absolute mismatches between supply and demand of more than 10% (table 3). These communities were located around conurbations in the North West of England and Greater London, with supply exceeding demand within cities, and demand exceeding supply in suburban communities (figure 3). In partition B, a supply-demand mismatch exceeding 10% is only observed for the surgical community on the south of the Thames Estuary, where demand exceeds supply by 25%, indicating a role for

| Table 3 Descriptive statistics for the three optimal partitions produced | | | |
|---|---|---|---|
| **Partition** | **A** | **B** | **C** |
| Communities (n) | 45 | 16 | 7 |
| Median number of cases per community | 78 998 (43 628–118 087) | 214 216 (122 823–314 022) | 574 403 (406 465–679 703) |
| Median number of treatment sites per community | 9 (5–17) | 25 (19–44) | 84 (56–98) |
| Absolute supply:demand mismatch (%) | 5.1 (2.9–10.2) | 4.1 (1.0–5.7) | 2.2 (1.0–2.9) |

IQRs are shown in parentheses where appropriate.

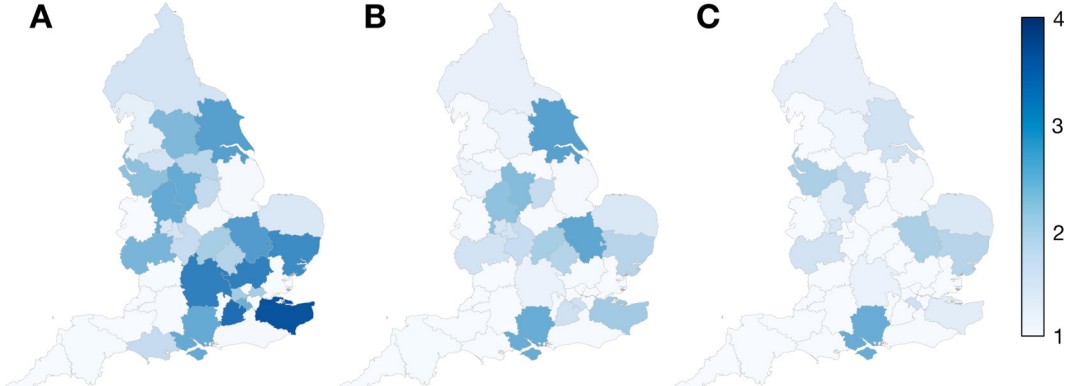

**Figure 2** The equivalent number of surgical communities active in each sustainability and transformation partnership (STP) as determined by the Equivalent Market Size (EMS) for configurations of 45 (panel A), 16 (panel B) and 7 (panel C) surgical communities. Areas of darker blue, 4 (eg, East Anglia), represent those areas with greatest difference between surgical communities and STPs, whereas lighter blue shows greater agreement (eg, Cornwall).

nearby surgical sites in East London which lie outside of the community. In partition C, the percentage difference between supply and demand does not exceed 5% in any community.

## DISCUSSION

Hospital providers, policymakers and clinicians urgently require solutions for managing the COVID-19 elective surgical aftershock. This describes a state where COVID-19 cases are in decline, in the context of strategically halted elective surgery and exponentially growing waiting-lists. The extraordinary levels of demand for operations now require radical new solutions to the way we organise and deliver surgical services. This study showed that there are existing hospital networks performing high volumes of low-risk procedures for low-risk, local patients. When we compare supply and demand for planned surgical care across England, the degree of mismatch varies widely, particularly around conurbations. Importantly, these data demonstrate that variation is reduced significantly when provider networks expand and smaller surgical communities coalesce into 16 larger geographic regions. We have

identified a large group of potentially eligible, fit, lower risk patients who could be asked to travel greater distances than the existing median of 13 km for their more minor surgery in order to shorten waiting times.

Central management of pooled waiting-lists across an increased number of both NHS and non-NHS providers offers an opportunity for greater collaboration between surgical centres and a better distribution of workload. It would provide enhanced system resilience in the context of future COVID-19 outbreaks to continue planned surgery in dedicated clean sites.[8 26 27] The scheme may have additional benefits including increased patient choice, greater workforce flexibility and maximisation of teams across areas, with increasing efficiency. There is a paucity of high-quality data on the effects of pooled waiting-lists.[28] Some evidence for their potential success has come from smaller, single-site initiatives piloting internal pooling of cases distributed to consultants in the same department.[29 30 30] Surgical pooling has been used successfully in crises to achieve waiting-list targets with work done by non-consultant grade surgeons and cases shifted to the private sector. Surgical pooling has also

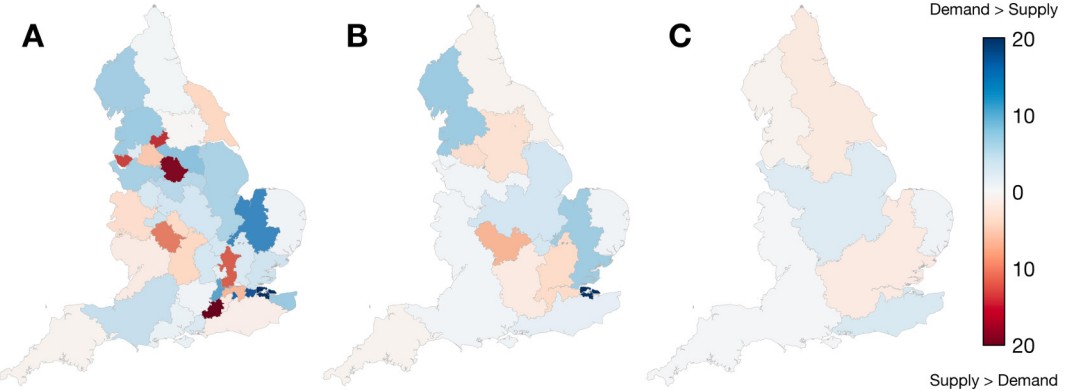

**Figure 3** The absolute percentage difference between the number of patients undergoing surgery resident within a surgical community (demand) and the number of procedures performed by hospitals within the community (supply) for configurations of 45 (panel A), 16 (panel B) and 7 (panel C) surgical communities. Areas in blue represent those surgical communities where procedures performed on patients outnumber those performed by the local providers.

been successful in matching existing supply to demand across transplant networks where donors are matched to recipients across larger regions, and sometimes between countries.[31] Greater choice and increased competition between providers for patients can be associated with reduced waiting times.[32] In this study, we remain agnostic as to the means by which providers within a pooled list community should collaborate, and accept that the timing and mechanism of any collaboration should reflect the idiosyncrasies of local contexts over time—and is a determination which is best made by local providers.

The London Patient Choice Project was set up to reduce long waiting times for patients awaiting ophthalmic and minor general surgery procedures. Waiting-lists were centrally pooled, managed and funded, with patients then given a choice on-site of care in order to obtain earlier treatment. This leads to a convergence of waiting times across providers by relieving those hospitals with longer lists.[13 33] Central purchasing of services was likely key to its success. On the strength of this pilot project, the English NHS undertook a national roll-out of patient choice, but without the central purchasing or coordination. 'Choose & Book' offered patients a choice of at least four hospitals which led some patients to attend a hospital other than the nearest one. Unpicking the effect of Choose & Book on waiting-lists separately to other initiatives piloted at the time is complex, but it is likely that the setting of targets and strong performance management were key drivers on reducing waiting times rather than patient choice alone.[34]

In the UK, patients generally favour the convenience and familiarity of a local provider. However, a MORI poll for the British Medical Association showed that if faced with a long wait, 27% of people would travel anywhere in the UK for treatment by the NHS.[35] Seventy-eight per cent of patients surveyed in the Isle of Wight were willing to travel to the mainland for elective surgery where the wait was shorter.[36] Greater patient travel has the potential to alleviate focal strain on services, but its practical application will require careful consideration. There are a number of barriers to travel—including patient mobility, age and risk as well as the cost of travel and the need for nearby family and friend support. In this study, selection of 'low-risk patients and procedures' acts to mitigate some of these concerns, although the identification of operative risk based on procedure, age and Charlson score may be limited, and clearly in practice a patient-specific, case-by-case approach would be required. Government subsidisation of travel would be an important intervention to reduce inequalities based on socioeconomic status, education level, vulnerability or social exclusion.[35] However, in the London pilot, there was no evidence of inequalities in uptake of the pooled list scheme by social class, educational attainment, income or ethnicity.[13] In the UK, with increasing centralisation of complex care, particularly cancer care, patients are often already asked to travel further.[37] The applicability of a pooled list surgical strategy varies according to the complexity of procedures and the need for in-person longitudinal follow-up with the operating centre. The finding that the majority of HVPs in this study are low-complexity procedures, with limited need for onward follow-up, supports the suitability of pooled provision for the HVPs identified.

In this study, we identified a degree of variation in the extent to which demand for planned surgery within a community is met by the capacity of hospitals located in the same community. This is in addition to the current variation in waiting-list lengths and COVID-19 infection and hospitalisation rates. If variability could be reduced, or eliminated, then capacity planning is simplified.[38] This strategy fits with NHS England's broader integration strategy as outlined in the Five Year Forward view and continued in the expansion of STPs to become larger integrated delivery systems.

The extent to which demand for surgical care will change as a result of COVID-19 remains uncertain. General practitioners and patients may prefer strategies of watchful waiting for minor surgical conditions, consequently reducing demand. Similarly, periods of lower community COVID-19 transmission may result in increased referral for surgical services before another wave of the pandemic takes hold. Regional variation in standardised rates of planned surgical procedures indicates that reductions in surgical demand may perhaps be greater in areas with lower pre-COVID-19 treatment thresholds and associated relative overuse.[39]

Similarly, the ability of hospitals to maintain pre-COVID-19 surgical capacity during and after the pandemic is uncertain. Recent research has demonstrated that some endoscopy departments in England maintained or even increased activity during the COVID-19 pandemic, while others stopped services entirely.[40] These findings indicate variation in local responses to the first wave of COVID-19 and allude to regional collaboration between surgical centres. In times of lower COVID-19 incidence, it could be expected that surgical supply may increase above pre-COVID-19 rates through provision of additional operating theatre capacity in evenings and weekends or the involvement of private sector care providers.[41] However, safely returning to baseline surgical capacity after a period of unprecedented disruption is a significant challenge in itself, and one where significant uncertainty remains. As a result of these uncertainties in future demand for, and supply of, surgical care, this study assumed future demand for planned surgical care would match historic demand from April 2017 to March 2018.

There are a number of limitations to the study. The COVID-19 epidemic is without precedent in recent history, so it was not possible to make substantially data-driven assumptions. The government has previously advised reducing national travel as a public health tool to limit COVID-19 spread.[42] While our model does encourage patient mobility and could be criticised for the risk of further spread, it also facilitates more effective regional strategies to dedicate sites as COVID-19 clean or dirty. Stringent infection control measures will be an essential part of any reintroduction of routine services. Currently, there is mounting evidence that patients are not seeking out routine care due to the perceived risk of COVID-19 infection.[43] There is therefore a possibility that patients

will choose not to undergo any elective procedures in the current climate, nor travel to an unknown hospital for that care. Pooled waiting-lists are often disliked by surgeons who site the lack of autonomy and patient ownership with an increased risk of misdiagnosis, unnecessary procedures listed and unaddressed patient complexities.[44 45] These risks can, and should, be mitigated by ensuring clear standardised patient pathways, patient triage and suitability assessments, clarity in the named responsible surgeon and pathways for ongoing continuity of care. Virtual platforms have become increasingly available during COVID-19 allowing remote consultation and triaging of patients prior to any procedures.[46]

This study included procedures of varying complexity and ability to increase surge capacity to overcome increased elective waiting-lists. Many of the most common procedures featured, including GI endoscopy, excisions of skin lesions and joint injection or aspiration, may be performed as 'day case' procedures and the ability to increase procedural throughput is less encumbered by the need for close anaesthetic support or high-dependency recovery space. In comparison, many higher risk procedures, including complex cardiac, cancer and orthopaedic surgery, are of lower volume. For example, during the study period, in England and Wales 16 000 planned colorectal cancer resections were performed, while 14 500 planned coronary artery bypass graft procedures were performed across the UK.[47 48] These procedures are more likely to require significant anaesthetic support, postoperative critical or high-dependency care and lead to longer inpatient stays. Planning to retain capacity for these complex procedures may therefore entail a different approach to the pooled list approach suggested for the HVPs identified in this study.

Additionally, in using historical surgical volume as a means of quantifying maximum capacity, the study does not incorporate measures to increase surge capacity above prior maximal volumes. As such, the maximal capacities identified for pooled list communities in this study may significantly underestimate the throughput which may be achieved with additional measures to support expansion of surgical capacity.

Finally, while we have identified a mismatch between current policy (STP boundaries) and practice (the natural networks of surgical providers), we appreciate that implementation of new integrated networks on a larger scale would require significant new resources and planning. A new system of funding flows, mechanisms for regional waiting-list coordination and a cost per case mechanism or other financial incentive would be required to support this new model.

The NHS, despite being centrally funded, functions as a disparate collection of separate providers with their own priorities and resource constraints. In the COVID-19 pandemic, pre-existing structures of service delivery within the NHS were temporarily transformed. Primary care providers collaborated at a regional level to provide COVID-19 care through a network of hubs while hospitals collaborated with one another to ensure some cancer care could continue at a smaller number of 'clean' hospital sites. As health systems across the world look to address an ever-growing backlog for planned care created by COVID-19, this trend of enhanced collaboration must continue. If the NHS is to overcome this backlog and cope with further waves of COVID-19, providers of surgical care must develop the means by which they may share a collective case load for low-risk patients. What is certain is that the NHS, along with most other healthcare delivery systems, is having to make seismic changes to the way it works in order to best manage ongoing complexities. This study provides a solution with greater regional capacity flexibility with which to respond and adapt. Redesigning arbitrary geographical boundaries to follow expanded natural surgical community networks has the potential to increase efficiency by flexing existing supply to meet demand. This, in addition to other key strategies, could have a profound effect on tackling the massive backlog of cases accruing during this deadly pandemic, thereby preventing further death, disability and reduced productivity from delayed surgery.

**Acknowledgements** Data management was provided by the Big Data and Analytical Unit (BDAU) at the Institute of Global Health Innovation (IGHI), Imperial College London.

**Collaborators** The PanSurg Collaborative: Amish Acharya, Ravi Aggarwal, Jonathan Clarke, Max Denning, James Kinross, Sheraz Markar, Guy Martin, Sam Mason, Sanjay Purkayastha, Alasdair Scott, Viknesh Sounderajah, Jasmine Winter Beatty and Seema Yalamanchili

**Contributors** JC and AM were involved in all aspects of the study. MB was involved in the development of the methodology and assisted in the formal analysis. SRM, MB and JK were involved in the planning, conduct and reporting of the study. JC has had access to all the data in the study and all authors had final responsibility for the decision to submit for publication.

**Funding** This article is an independent research supported by grants from the Peter Sowerby Foundation and the National Institute for Health Research (NIHR) Imperial Patient Safety and Translational Research Centre (PSTRC_2016_004). Infrastructure support for this work was provided by the NIHR Imperial Biomedical Research Centre (BRC 1215-20013). MB and JC acknowledge the support from EPSRC grant EP/N014529/1 supporting the EPSRC Centre for Mathematics of Precision Healthcare. JC acknowledges support from the Wellcome Trust (215938/Z/19/Z). JK reports funding from H2020-ITN grant, NIHR-i4i and CRUK. Johnson & Johnson has supported the activities of the PanSurg Collaborative with an educational grant.

**Competing interests** JK reports consultancy for Verb robotics/Ethicon and is a shareholder of OneWelbeck Day Surgery.

**Patient consent for publication** Not required.

**Ethics approval** This study received local ethical approval through the Imperial College Research Ethics Committee (17IC4178).

**Data availability statement** Data used in this study were obtained from NHS Digital for the purpose of this work and may only be accessed through direct application to NHS Digital. Patient-level data are required for the analyses conducted, and therefore sharing of data pertaining to this study is not possible. Data for surgical community assignments are available from the authors on request.

**ORCID iDs**
Jonathan Clarke http://orcid.org/0000-0003-1495-7746
Sheraz Rehan Markar http://orcid.org/0000-0001-8650-2017
Mauricio Barahona http://orcid.org/0000-0002-1089-5675

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
