## [Reviewer comments · BMJ Open]

ARTICLE DETAILS

TITLE (PROVISIONAL)	A New Geographic Model of Care to Manage the Post-COVID-19 Elective Surgery Aftershock in England: A Retrospective Observational Study
AUTHORS	Clarke, Jonathan; Murray, Alice; Markar, Sheraz; Barahona, Mauricio; Kinross, James

VERSION 1 – REVIEW

REVIEWER	John Hick Hennepin Healthcare, University of Minnesota Minneapolis, MN USA
REVIEW RETURNED	01-Aug-2020

GENERAL COMMENTS	The authors are to be commended for tackling a key logistical problem facing healthcare systems during COVID-19. Though the solutions may be most applicable to national healthcare systems they offer an interesting perspective on fully utilizing the resources in a given regional geographic area. I have a few concerns that I would like to see addressed prior to publication. As I understand it, the available 'delta' between supply and demand was calculated using the total procedures done minus the total procedures done on local residents. Given the nature of potential referrals, I am not surprised that the Greater London area was thus calculated to have more 'supply' but am concerned that this actually reflects more likely travel to that area for procedures, and that the actual available surgical capacity is significantly greater - given that physicians in smaller hospitals may have only half-day schedules of procedures not detected by the current methods the potential to significantly increase capacity may exist. Further surge capacity may also be possible that is not available by the data because those plans were not activated. Diagnostic endoscopies in general do not qualify as 'surgery' and could be conducted regardless of the impact on hospitals of COVID since they rarely result in hospital overnight stay and therefore do not compete with the inpatient capacity of the hospital - therefore 'catch-up' on those procedures could be scheduled regionally without difficult - orthopedic procedures that might require an inpatient stay however might be dependent on available beds which could be problematic if there is a COVID-19 surge at the time. Still, the authors have confirmed some of the potential offered by the pilot trial in London to maximize use of procedural capacity across healthcare systems and proposed what seems to be a reasonable regional structure that is worth more detailed investigation to vet the assumptions about surgical capacity. A final recommendation might be about developing a prioritization scheme based on the urgency of the elective procedure. Common procedures could be
--

	divided by specialists and blocks of time / resources and based on the hospital capacity in the area at the time, electives requiring inpatient stays could be throttled up or down depending on capacity of the hospitals relative to COVID while maintaining a steady volume of procedures such as diagnostic endoscopy. Thank you for the opportunity to review this interesting manuscript!
--	--

REVIEWER	William B Weeks, MD, PhD, MBA Microsoft, USA
REVIEW RETURNED	12-Aug-2020

GENERAL COMMENTS	This is an interesting paper that uses a retrospective analysis to consider a forward-looking approach to 'catching up' on elective surgeries in the UK. I have a couple of concerns about the paper. The first, perhaps minor, one regards the numbers. The first operative day is defined and the analysis is limited to procedures done in the first operative day (which suggests to me, those in one operation). The challenge that I see is that 4.3 million patients had 5.7 million admissions with 7.8 million interventions. While I guess that each admission had about 1.4 procedures, I think it is likely that those 1.4 procedures were really just one surgery (i.e., a 'knee arthroscopy' might include several procedures, including installation of the prosthesis, but also, for instance, removal of the old knee). The key is to make the interventions meaningful for the purposes of allocation to different hospitals. Perhaps the authors did that, it's just not clear to me. Presumptively, a number of patients had >1 admission (which makes sense, a lot of times one hip is done, followed by rehab, and followed by the 2nd hip). The bigger issue that I have is that the authors don't consider overuse of elective surgeries. The king's fund did a nice study of reconfiguring elective surgical care (https://www.kingsfund.org.uk/publications/reconfiguration-clinical-services/elective-surgical) and of geographic variation in elective surgeries (https://www.kingsfund.org.uk/sites/default/files/field/field_publication_file/Variations-in-health-care-good-bad-inexplicable-report-The-Kings-Fund-April-2011.pdf). While it is likely that there is pent-up demand for elective surgeries (though describing that as 'exponential' is perhaps going too far), it is possible that pent-up demand is highest in NHS areas where there is relative overuse of elective surgeries (likely due to lower treatment thresholds). One might think that NHS might mitigate health services overuse by providing, say, shared decision-making tools which might reduce demand. Nonetheless, the authors don't really consider this aspect of elective care (admittedly, the lower GI series are likely recommended screening and are a big part of the demand). But I'd like to see some kind of recognition of geographic variation and an inclusion (maybe in a sensitivity analysis) of reducing overuse as an additional mechanism to reduce pent-up demand. The only other 2 issues I have are that the citations are variable before or after punctuation. They should be after. And, secondly, the authors use the word 'supply' as though it were fixed. Just as demand might rise and fall, so might supply - indeed, a surgeon might increase the number of cases he does each day to rapidly increase supply. Supply is not fixed.
---

VERSION 1 – AUTHOR RESPONSE

Reviewer: 1

Reviewer Name: John Hick Institution
and Country:

Hennepin Healthcare, University of Minnesota Minneapolis, MN USA Competing interests: None declared

Please leave your comments for the authors below

The authors are to be commended for tackling a key logistical problem facing healthcare systems during COVID-19. Though the solutions may be most applicable to national healthcare systems they offer an interesting perspective on fully utilizing the resources in a given regional geographic area. I have a few concerns that I would like to see addressed prior to publication.

As I understand it, the available 'delta' between supply and demand was calculated using the total procedures done minus the total procedures done on local residents. Given the nature of potential referrals, I am not surprised that the Greater London area was thus calculated to have more 'supply' but am concerned that this actually reflects more likely travel to that area for procedures, and that the actual available surgical capacity is significantly greater - given that physicians in smaller hospitals may have only half-day schedules of procedures not detected by the current methods the potential to significantly increase capacity may exist. Further surge capacity may also be possible that is not available by the data because those plans were not activated.

Thank you for highlighting this key point. In this study we decided to fix supply at retrospective baseline figures, as there is significant uncertainty relating to the ability of individual organisations to expand their capacity or maintain baseline capacity should COVID-related demand for care increase again. We have therefore inserted this into the discussion to explain our reasoning. In the NHS generally, under 'normal' conditions, operating theatre capacity is close to full, and thereby represents a logistical ceiling to the number of cases that can be performed. There may be the option to increase surgical capacity through leveraging private sector provider theatre space, and this is something we now address in the discussion.

Diagnostic endoscopies in general do not qualify as 'surgery' and could be conducted regardless of the impact on hospitals of COVID since they rarely result in hospital overnight stay and therefore do not compete with the inpatient capacity of the hospital - therefore 'catch-up' on those procedures could be scheduled regionally without difficulty - orthopedic procedures that might require an inpatient stay however might be dependent on available beds which could be problematic if there is a COVID-19 surge at the time.

Still, the authors have confirmed some of the potential offered by the pilot trial in London to maximize use of procedural capacity across healthcare systems and proposed what seems to be a reasonable regional structure that is worth more detailed investigation to vet the assumptions about surgical capacity.

We absolutely agree that endoscopy may be managed differently to other common day case and inpatient procedures. We were keen to include a wide variety of procedures in this study with the intention to capture as much of everyday planned surgical demand as possible. Patients undergoing endoscopy, despite being day case procedures, will still be subject to stringent Covid testing and precautions. In addition, due to the significant backlog of patients, particularly those awaiting screening, there will still need to be a flexibility in how and importantly, where, these patients are managed. As you say, it is easier to 'catch-up' for some procedures than others. This is important and is something with have added to the discussion following your suggestion.

A final recommendation might be about developing a prioritization scheme based on the urgency of the elective procedure. Common procedures could be divided by specialists and blocks of time / resources and based on the hospital capacity in the area at the time, electives requiring inpatient stays could be throttled up or down depending on capacity of the hospitals relative to COVID while maintaining a steady volume of procedures such as diagnostic endoscopy. Thank you for the opportunity to review this interesting manuscript!

We are in absolute agreement with you. It is a key part of the model that each regional surgical community identified has its own idiosyncratic modes of care delivery and challenges to 'catching up' after COVID. In this study we tried to remain agnostic as to how localities should overcome their own backlogs, beyond asserting that a degree of local collaboration is possible, and likely to enhance system-wide resilience. Following your comment we have added to the discussion to clarify that a range of solutions may be adopted that best fit local demands and capabilities. Thank you.

Reviewer: 2

Reviewer Name: William B Weeks, MD, PhD, MBA Institution and Country: Microsoft, USA Competing interests: None declared

Please leave your comments for the authors below

This is an interesting paper that uses a retrospective analysis to consider a forward-looking approach to 'catching up' on elective surgeries in the UK.

I have a couple of concerns about the paper.

The first, perhaps minor, one regards the numbers. The first operative day is defined and the analysis is limited to procedures done in the first operative day (which suggests to me, those in one operation). The challenge that I see is that 4.3 million patients had 5.7 million admissions with 7.8 million interventions. While I guess that each admission had about 1.4 procedures, I think it is likely that those 1.4 procedures were really just one surgery (i.e., a 'knee arthroscopy' might include several

procedures, including installation of the prosthesis, but also, for instance, removal of the old knee). The key is to make the interventions meaningful for the purposes of allocation to different hospitals. Perhaps the authors did that, it's just not clear to me. Presumptively, a number of patients had >1 admission (which makes sense, a lot of times one hip is done, followed by rehab, and followed by the 2nd hip).

Thank you, this is an important point that we have now re-addresses in the manuscript. We took only the procedures occurring on the first operative day of an admission in order to exclude double counting post-operative complications requiring returns to theatre on days after the first operative day. It is not possible in the general case to differentiate multi-stage, single admission procedures from unplanned returns to theatre in Hospital Episode Statistics, therefore only the procedures performed on the first operative day were counted.

You are absolutely correct in the finding of multiple procedures for each admission. This is as a result of our coding practices in England and multiple different procedures being undertaken within the same operating theatre session. In some cases the pair of procedures may be closely linked, as in the example of knee surgery, or in the case of a laparotomy and a colonic resection. In other cases, such as laparoscopic cholecystectomy and an inguinal hernia repair occurring together, the difference is more distinct. In this study we wanted to capture as much of the variety of surgical procedures as possible. Similarly, identifying the 'dominant' procedure amongst multi-procedure operations is highly subjective. We have added a comment to our methods section to clarify this.

The bigger issue that I have is that the authors don't consider overuse of elective surgeries. The king's fund did a nice study of reconfiguring elective surgical care (<https://www.kingsfund.org.uk/publications/reconfigurationclinical-services/elective-surgical>) and of geographic variation in elective surgeries

(https://www.kingsfund.org.uk/sites/default/files/field/field_publication_file/Variations-in-health-care-goodbad-inexplicable-report-The-Kings-Fund-April-2011.pdf). While it is likely that there is pent-up demand for elective surgeries (though describing that as 'exponential' is perhaps going too far), it is possible that pent-up demand is highest in NHS areas where there is relative overuse of elective surgeries (likely due to lower treatment thresholds). One might think that NHS might mitigate health services overuse by providing, say, shared decision-making tools which might reduce demand. Nonetheless, the authors don't really consider this aspect of elective care (admittedly, the lower GI series are likely recommended screening and are a big part of the demand). But I'd like to see some kind of recognition of geographic variation and an inclusion (maybe in a sensitivity analysis) of reducing overuse as an additional mechanism to reduce pent-up demand.

Thank you for this very reasonable suggestion. We have amended the discussion to include reference to both regional variation in treatment thresholds and the likely attenuation of demand following Covid-19. We agree that there are a number of ways of tackling backlog, and certainly "reducing demand" or focusing on only "high value" procedures is one way. Shared decision making is key to ensuring appropriateness of intervention in these cases, although the evidence for SDM and its effect on utilisation can be inconsistent. While there is great variation in utilisation as you say, the NHS has made significant changes to Commissioning what might be considered preference- sensitive procedures through initiatives like "choose wisely" and also more stringent qualifications for surgery as is in the case of hernia repair. Estimating the extent to which demand will be reduced post-Covid is likely to vary between patients, procedures, across geographies and over time, and its evaluation would be an important and exciting study in its own right. Here we make the assumption of demand for elective surgical care remaining unchanged as we can't confidently support any deviation from this

based on available data. We have added comment in the discussion to support and explain this interesting part of the debate.

The only other 2 issues I have are that the citations are variable before or after punctuation. They should be after. And, secondly, the authors use the word 'supply' as though it were fixed. Just as demand might rise and fall, so might supply - indeed, a surgeon might increase the number of cases he does each day to rapidly increase supply. Supply is not fixed.

Thank you. We have amended the references as suggested. We have additionally inserted a section in the discussion to reflect variation in supply as well as demand.

VERSION 2 – REVIEW

REVIEWER	John Hick Hennepin Healthcare, University of Minnesota – USA
REVIEW RETURNED	08-Sep-2020

GENERAL COMMENTS	the authors have tackled a key question arising from the cessation / postponement of many procedures due to COVID-19. they have made revisions which are appreciated. their limitations should still acknowledge that GI endoscopy does not carry the same considerations for impact on hospital surge capacity that cardiac and major orthopedic procedures do (some additional information about the relative frequency of those procedures would be nice). further, the methods are not able to capture surgical and procedural 'surge capacity' and rely on a historical assumption of 'maximal capacity' which may significantly under-estimate actual regional capacity. the study is an interesting look at the math behind the likely delays to 'catch up' on procedures and potential 'load-balancing' strategies.
---

REVIEWER	William B Weeks Microsoft Research, USA
REVIEW RETURNED	02-Sep-2020

GENERAL COMMENTS	The authors have responded well to my concerns and I have no others.
--

VERSION 2 – AUTHOR RESPONSE

Reviewer: 1

Reviewer Name: John Hick

The authors have tackled a key question arising from the cessation / postponement of many procedures due to COVID-19. they have made revisions which are appreciated.

Thank you for your helpful comments that have certainly improved the manuscript.

Their limitations should still acknowledge that GI endoscopy does not carry the same considerations for impact on hospital surge capacity that cardiac and major orthopedic procedures do (some additional information about the relative frequency of those procedures would be nice).

Thank you. We have added a paragraph to the discussion which addresses the difference between endoscopy and more complex procedures with a view to expansion of capacity. To provide context, we have included annual volumes for the same period for colorectal cancer resection in England and Wales and for coronary artery bypass grafting in the UK from contemporaneous national audits (<https://www.nicor.org.uk/wp-content/uploads/2019/09/ACS-2019-Summary-Report-final.pdf> and <https://www.nboca.org.uk/content/uploads/2020/01/NBOCA-2019-V2.0.pdf>). These are the most reliable estimates by which to contextualise volumes.

Further, the methods are not able to capture surgical and procedural 'surge capacity' and rely on a historical assumption of 'maximal capacity' which may significantly under-estimate actual regional capacity.

Thank you. We have added a short paragraph to the discussion to explicitly address this important point.

the study is an interesting look at the math behind the likely delays to 'catch up' on procedures and potential 'load-balancing' strategies.

Reviewer: 2

Reviewer Name: William B Weeks

The authors have responded well to my concerns and I have no others.

Thank you for your helpful comments that have certainly improved the manuscript.